# The Role of SARS-CoV-2 Nucleocapsidic Antigen and Krebs von den Lungen 6 Serum Levels in Predicting COVID-19 Pneumonia Outcome

**DOI:** 10.3390/diagnostics14060642

**Published:** 2024-03-18

**Authors:** Stefano Sanduzzi Zamparelli, Vincenzo Fucci, Gaetano Rea, Francesco Perna, Marialuisa Bocchino, Alessandro Sanduzzi Zamparelli

**Affiliations:** 1Division of Pneumology, A. Cardarelli Hospital, 80131 Naples, Italy; 2Department of Clinical Medicine and Surgery, Section of Respiratory Diseases, University Federico II, Azienda Ospedaliera dei Colli-Monaldi Hospital, 80131 Naples, Italy; vincenzofucci87@gmail.com (V.F.); federico90@libero.it (F.P.); marialuisa.bocchino@unina.it (M.B.); sanduzzi@unina.it (A.S.Z.); 3Department of Radiology, Azienda Ospedaliera dei Colli-Monaldi Hospital, 80131 Naples, Italy; gaetano.rea71@gmail.com; 4Staff of UNESCO Chair for Health Education and Sustainable Development, University Federico II, 80131 Naples, Italy

**Keywords:** COVID-19 pneumonia, SARS-CoV-2 nucleocapsid antigen, Krebs von del Lungen-6

## Abstract

Background: The COVID-19 pandemic caused by SARS-CoV-2 continues to pose a significant threat worldwide, with severe cases leading to hospitalization and death. This study aims to evaluate the potential use of serum nucleocapsid antigen (NAg) and Krebs von den Lungen-6 glycoprotein (KL-6) as biomarkers of severe COVID-19 and to investigate their correlation with clinical, radiological, and biochemical parameters. Methods: This retrospective study included 128 patients with confirmed SARS-CoV-2 infection admitted to a Neapolitan hospital in Italy between October 2020 and July 2021. Demographic, clinical, and laboratory data were collected, including serum levels of NAg and KL-6. The Chung et al. Computed Tomography Severity Score (TSS) was used to assess the severity of pneumonia, and outcomes were classified as home discharge, rehabilitation, and death. Statistical analyses were performed to compare Group I (home discharge and rehabilitation) and Group II (death, sub-intensive care, and ICU stay) based on demographic data, laboratory parameters, and TSS. Results: Group II patients showed worse outcomes with higher levels of NAg, KL-6, and inflammatory markers, including interleukin-6 (IL-6), interleukin-2 receptor (IL-2R), and adrenomedullin. TSS was also significantly higher in Group II, with a positive correlation between TSS and NAg and KL-6 levels. Group I patients had higher values of hemoglobin (Hb) and platelets (PLT), while Group II patients had higher values of C-reactive protein (CRP), procalcitonin (PCT), D-Dimer, and glycemia. No significant difference was observed in gender distribution. Conclusions: Serum NAg and KL-6 levels are potential biomarkers of severe COVID-19 pneumonia, with higher levels indicating greater inflammation and organ damage. NAg may help identify infected patients at an increased risk of severe COVID-19 and ensure their admission to the most appropriate level of care. KL-6 may help predict interstitial lung damage and the severity of clinical features. Further studies are needed to establish a decision-making cut-off for these biomarkers in COVID-19.

## 1. Introduction

The COVID-19 syndrome caused by SARS-CoV-2 results in a wide range of clinical manifestations varying from mild to critical cases. Severe and critical patients may experience potentially life-threatening complications, including respiratory distress syndrome, hyperinflammatory and hypercoagulable status due to cytokine storm syndrome, and multiorgan failure [1,2,3]. However, predicting the risk of intubation or death remains challenging due to the lack of effective biomarkers [4,5]. Although significant advances have been made in the therapeutic management of COVID-19 by introducing antiviral agents, neutralizing antibody therapies, Janus kinase inhibitors, steroids, or natriuretics, several important questions remain regarding its treatment [6,7]. 

Based on the above, recently, several biochemical biomarkers, including Krebs von den Lungen-6 (KL-6), C reactive protein (CRP), interleukin-6 (IL-6), D-dimer, and neutrophil/lymphocyte ratio (NLR), have been analyzed to determine their predictive value [8,9]. 

Recent studies [10,11] suggest that the SARS-CoV-2 Nucleocapsidic protein (Nag) could be a reliable marker for early detection of COVID-19. This is due to its identification in gargles and nasopharyngeal swabs or its quantitative measurement in patients’ serum/plasma.

The NAg, together with the spike glycoprotein (S), envelope (E), and membrane protein (M), constitutes one of the four essential structural proteins of SARS-CoV-2 [12].

Upon entering the host cell, thanks to the binding between angiotensin-converting enzyme II (ACE 2) and S protein, the NAg dissociates itself from the positive strand (+) RNA genome and helps to regulate viral transcription by enveloping positive-strand RNA and forming a viral replication-transcriptional complex (RTC) with SARS-CoV-2 nonstructural proteins [13]. Additionally, the NAg inhibits an inherent antiviral immune defense mechanism called RNA interference (RNAi), preventing the recognition and cleavage of viral dsRNA in host cells, thus regulating viral host cell cycle progression, replication, and SARS-CoV-2 gene expression program [14,15].

KL-6 is a glycoprotein expressed by damaged type II alveolar epithelial cells via shedding of the extracellular MUCIN 1 domain. KL-6 comprises a stretch of 20 amino acids rich in glycosylated serine and threonine residue, forming a rigid structure that provides a barrier against microbial and viral attacks. Notably, the release of KL-6 in the serum is facilitated by ADAM17, a metalloprotease responsible for the cleavage of ACE2, the primary receptor of SARS-CoV-2 [16,17]. Despite its anti-inflammatory properties and ability to inhibit Toll-like receptor (TLR) signaling, KL-6 has also been associated with the development of fibrosis and the progression of lung cancer [18]. Although initially identified as a tumor marker, subsequent research has found that KL-6 may promote fibroblast migration and inhibit cell–cell adhesion, leading to its identification as a marker of interstitial lung fibrosis [19].

A wide range of studies [20,21,22] indicates a significant increase in KL-6 levels among COVID-19 patients, strongly correlated with lung function as assessed by lung ultrasound (LUS). Moreover, this marker seems helpful in predicting the need for intubation and is highly associated with mortality. 

For this reason, in our study, we sought to characterize the role of SARS-CoV-2 NAg and KL-6 in a subset of patients admitted to our hospital that progressed to severe conditions. Our objective was to examine the potential correlation between its levels and lung function in severe and critical COVID-19 patients to predict the need for intubation and mortality rates.

## 2. Materials and Methods

The study included COVID-19-related pneumonia patients consecutively hospitalized in various care settings at “Azienda Ospedaliera dei Colli Monaldi-Cotugno-CTO” in Naples, Italy, from October 2020 to July 2021. All participants provided written informed consent before participating in the study based on the principles of the Declaration of Helsinki. The study protocol and amendments were approved by the institutional review board and independent ethics committees of Azienda Ospedaliera dei Colli Monaldi-Cotugno-CTO (local ethical committee: ref AOC/0013770/2020); the study was conducted according to Good Clinical Practice guidelines defined by the International Council for Harmonisation and local laws. Demographic, clinical, radiological, and laboratory data were extracted from electronic medical records. To be included in the study, patients were required to be aged at least 18 years old and have a positive RT-PCR nasopharyngeal swab at hospital admittance and detectable serum levels of SARS-CoV-2 NAg > 1 pg/mL. Patients were excluded if they were over 90 years old, had no available RT-PCR nasopharyngeal swab at admittance, or had undetectable SARS-CoV-2 NAg serum levels (<1 pg/mL). Clinical, radiological, and laboratory data were evaluated upon hospital admission and weekly until home discharge, rehabilitation, or in-hospital death. Database creation ended in June 2022. The diagnosis of COVID-19 pneumonia was established by evaluating the clinical symptoms and subsequently confirming the presence of SARS-CoV-2 RNA in nasopharyngeal swabs utilizing a SARS-CoV-2 nucleic acid detection kit in accordance with the protocol specified by the manufacturer, Anatolia Gene Work. The Lumipulse^®^ G SARS-CoV-2 Antigen Chemiluminescent Enzyme Immunoassay (CLEIA) was used to measure the SARS-CoV-2 NAg in the sera of COVID-19 patients. To perform the test, 100 μL of serum was added to the anti-SARS-CoV-2 antigen monoclonal antibody-coated magnetic particle solution and incubated for 10 min at 37 °C. The solution was then washed, and alkaline phosphatase-conjugated anti-SARS-CoV-2 antigen monoclonal antibody was added, followed by another 10 min incubation at 37 °C. After a second wash, substrate solution was added and incubated for 5 min at 37 °C. The fully automated LUMIPULSE^®^ G600II e G1200 (Fujirebio, Tokyo, Japan) was used to perform the assay. In a previous study [23], we demonstrated the efficacy of the Lumipulse^®^ test for serum utilization by examining the correlation between SARS-CoV-2 NAg and RNA serum levels. Evaluation of biochemical and inflammatory parameters including albumin, sodium, potassium, creatinine, glucose, urea, CRP, procalcitonin, ferritin, IL-6, interleukin-2 receptor (IL-2R), D-dimer, aspartate aminotransferase (AST), and alanine aminotransferase (ALT) were determined using the AtellicaTM CH analyzer according to the manufacturer’s protocol. The serum concentration of KL-6 was determined using a KL-6 antibody kit (LUMIPULSE, Fujirebio, Tokyo, Japan) in accordance with the manufacturer’s protocol, utilizing the same method as for the SARS-CoV-2 NAg assay. A computed tomography (CT) scan was used to examine any pathological involvement of the lungs. To evaluate the severity of pneumonia, the Chung et al. Computed Tomography Severity Score (TSS) was used [24]. Each lobe of the lung is given a score from 0 to 4, based on the percentage of lung involvement. The scores are as follows: 0 (no involvement), 1 (1–25% involvement), 2 (26–50% involvement), 3 (51–75% involvement), and 4 (76–100% involvement). The total severity score (TSS) is the sum of the individual lobe scores, ranging from 0 to 20. The patients were divided into two groups for statistical analysis. Group I (GI) included patients who were discharged to go home or to rehabilitation therapy, while Group II (GII) included patients who either passed away or required sub-intensive care or ICU stay. Demographic data and laboratory parameters, including serum SARS-CoV-2 NAg and KL-6, were compared between the two groups using the Mann–Whitney U-test. Gender distribution was evaluated using the χ^2^ test. The Mann–Whitney U-test also compared the TSS of Group I and Group II. Pearson correlation was performed to assess the relationship between NAg, KL-6, TSS, and other parameters. Prevalence, odds ratio, and χ^2^ test were used to evaluate the differences in complications between GI and GII.

## 3. Results

The study included 128 subjects with a mean age of 64.75 years (ranging from 22 to 88 years); 92 (71.87%) subjects were male, and 36 (28.12%) were female. According to the previous division, GI comprised 76 subjects (59.37%), while GII comprised 52 subjects (40.62%). The mean age of GI patients was 60.56 years, and that of GII patients was 71.26 years. The Mann–Whitney U-test confirmed that GII had a statistically significant greater mean age. Male subjects accounted for 76.32% of GI patients (58/76), while 23.68% (18/76) were female. In GII, 65.38% (34/52) were male, and 34.62% (18/52) were female. No significant difference was reported between the two groups using the χ^2^ test (Table 1).

The Mann–Whitney U-test was used to compare the means between GI and GII. The results showed that among blood count parameters, GI had statistically significantly (*p* > 0.05) higher values of red blood cells (RBC), hemoglobin (Hb), monocytes, eosinophils, and platelets (PLT). On the other hand, Group II had statistically significantly (*p* > 0.05) higher neutrophil values. No difference was observed for lymphocytes and basophils (Table 2).

The study found that among the biochemical and serological parameters, GI had statistically significantly (*p* > 0.05) higher values for ALT, 25-hydroxyvitamin (OH) D_3_, albumin, total proteins, calcium, and corrected calcium. On the other hand, Group II had statistically significantly (*p* > 0.05) higher values for creatinine, azotemia, potassium, and glycemia. No difference was recorded for aspartate aminotransferase (AST) and sodium (Table 3).

GI showed statistically significantly higher values (*p* > 0.05) of ferritin among the inflammation parameters and cytokines. On the other hand, GII showed statistically significantly higher values (*p* > 0.05) of RCP, procalcitonin (PCT), erythrocyte sedimentation rate (ESR), D-Dimer, IL-6, Interleukin-2 Receptor (IL-2R), and adrenomedullin. No difference was recorded for fibrinogen (Table 4).

### 3.1. SARS-CoV-2 Nucleocapsidic Antigen

The concentration of NAg was measured in all patients upon admission to the hospital. The mean values of GI and GII were compared using the Mann–Whitney U-test. Patients in GII, who had worse outcomes, had statistically significantly higher values (μ_II_ = 400,77 pg/mL). In our study population, NAg levels had a statistically significant positive correlation, as determined by the Pearson test, with CRP, PCT, TSS, KL-6, and IL-6 values. However, no correlation was found with IL-2R levels (Table 4 and Table 5, Figure 1). Despite the modest correlation between Nag and the primary laboratory and/or radiological biomarkers of inflammation, it is noteworthy that there exists a positive correlation between NAg and KL-6. These two biomarkers have shown potential in assessing the severity of lung involvement.

### 3.2. Krebs von den Lungen-6

KL-6 concentration was measured at hospital admission in all patients, and the mean values of GI and GII were compared using the Mann–Whitney U-test. Patients in GII, who had worse outcomes, had statistically significant higher values (μ_II_ = 1159 U/mL). KL-6 levels in our study population were found to have a statistically significant positive correlation with NAg and TSS values, as verified by the Pearson test. However, no correlation was found between CRP and IL-6 (Table 4 and Table 5, Figure 2).

### 3.3. Total Severity Score

In this study, the TSS value for hospital admittance was evaluated for all patients, and the mean values of GI and GII were compared using the Mann–Whitney U-test. The patients in GII, who had worse outcomes, showed significantly higher values (μ_II_ = 15). Furthermore, the TSS of our study population exhibited a statistically significant positive correlation, verified by the Pearson test, with NAg, CRP, KL-6, and IL-6 (Table 4 and Table 5, Figure 3).

**Table 5 diagnostics-14-00642-t005:** Correlation between KL-6, NAg, TSS, and disease severity markers.

Disease Severity Markers	NAg		KL-6		TSS	
	r	*p*-Value	r	*p*-Value	r	*p*-Value
CRP	+0.211	0.01 *	+0.104	0.43	+0.250	0.005 *
PCT	+0.207	0.02 *	NR	NR	NR	NR
TSS	+0.199	0.02 *	+0.268	0.04 *	NA	NA
KL-6	+0.278	0.01 *	NR	NR	+0.268	0.04 *
IL-6	+0.248	0.005 *	+0.129	0.33	+0.248	0.005 *
IL-2R	+0.145	0.1 *	NR	NR	NR	NR
NAg	NA	NA	+0.278	0.03 *	+0.199	0.02 *

NAg: nucleocapsidic antigen; KL-6: Krebs von den Lungen-6; r: Pearson correlation; CRP: C reactive protein; PCT: procalcitonin; TSS: total severity score; IL-6: interleukin-6; IL-2R: interleukin-2 receptor; NR: not relevant; NA: not applicable; *: statistically significant (*p* > 0.05).

**Figure 1 diagnostics-14-00642-f001:**
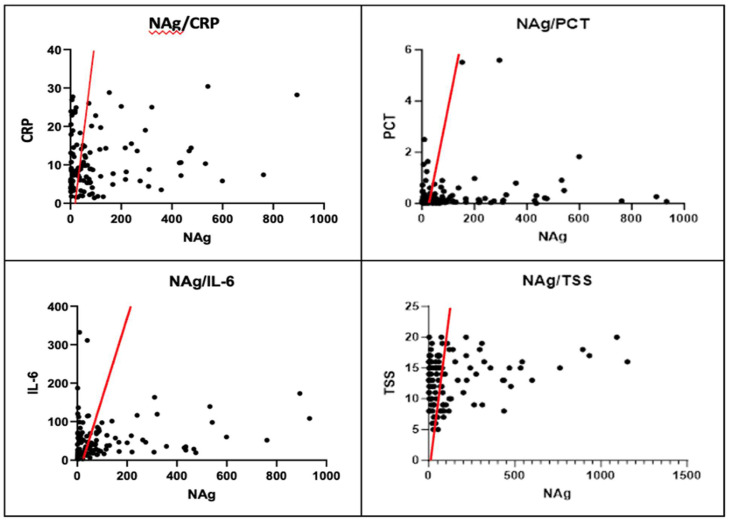
Positive Pearson correlation between NAg levels and CRP, PCT, IL-6, and TSS values.

**Figure 2 diagnostics-14-00642-f002:**
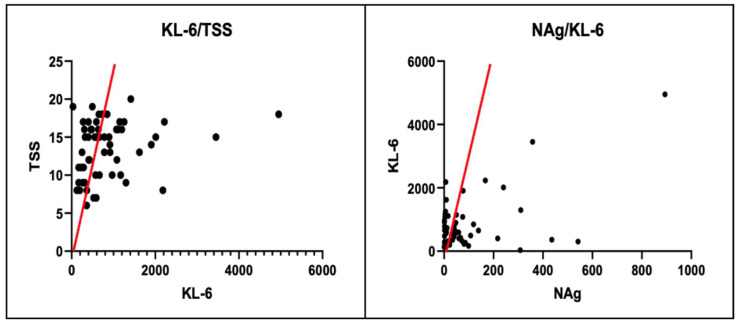
Positive Pearson correlation between KL-6 levels and NAg, TSS values.

**Figure 3 diagnostics-14-00642-f003:**
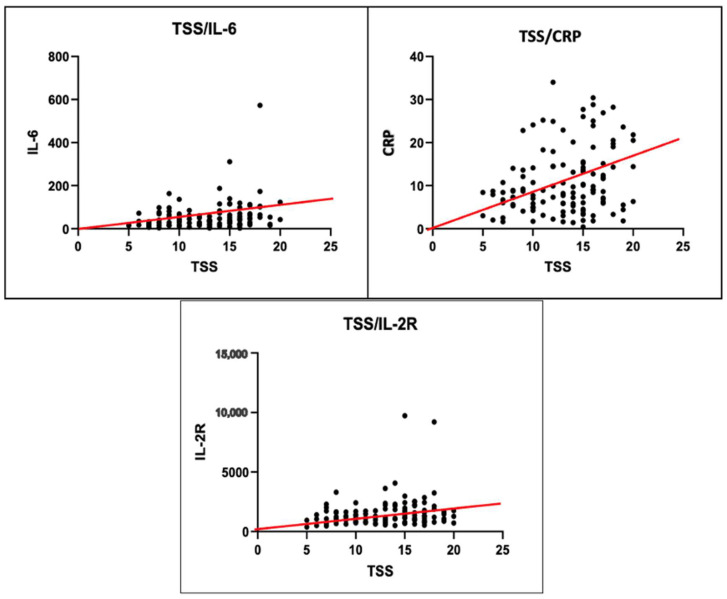
Positive Pearson correlation between TSS levels and IL-6, CRP, and IL-2R values.

### 3.4. Complications

Several complications were observed in our study population throughout the hospitalization period. Prevalence and odds ratio (OR) of anemia, renal failure, and disiony were estimated for the entire population, GI, and GII. The OR was verified using the χ^2^ test (Table 6).

## 4. Discussion

Patients with the worst outcomes in our study population showed a higher mean age (μ_II_ = 71.26). These data are in line with the literature, which shows a higher frequency of diseases and ICU stays in the elderly population [25,26,27,28]. Many studies evidenced an association between male sex and unfavorable outcomes. Nevertheless, our study did not verify this association (Table 1) [29,30].

Among blood count parameters, GI showed higher values of Hb and PLT: those data suggest that lower disease severity is associated with a lower prevalence of complications such as bleeding, thrombosis, and anemia (Table 2 and Table 6).

Immunity dysregulation and severe inflammation are ascertained COVID-19 features leading to morbidity and mortality due to organ damage [31,32]. Among inflammation markers and cytokines examined, GII showed higher values of CRP, PCT, D-Dimer, IL-6, IL-2R, and adrenomedullin, confirming a higher inflammatory status in patients with more severe outcomes (Table 4). IL-6 is a proinflammatory cytokine related to disease severity and unfavorable outcomes such as ICU stay, ARDS, and severe disease; as a matter of fact, monoclonal antibody Tocilizumab is commonly prescribed to patients affected with the most severe forms of COVID-19 (Table 4) [33,34,35,36]. IL-2R is known as a T-lymphocyte and fibroblast activation marker. The literature shows that its levels are increased in ARDS and patients with unfavorable outcomes. In our study population, GII patients showed higher levels than GI (Table 4) [32,37,38]. PCT is primarily undetectable in viral infections while commonly elevated in bacterial and mycotic ones. Our study population presented higher PCT values than reference values in both GI and GII, although GII resulted in a statistically significant higher mean value. These data confirm how most severe outcomes of COVID-19 are related to more frequent complications such as sepsis and bacterial pulmonary superinfection (Table 4) [39,40]. D-dimer increase is an expression of a procoagulative status and is linked to high mortality and ICU admission in patients with community and hospital-associated pneumonia; on the other hand, low levels are associated with a low risk of complications (Table 4).

Some studies highlighted that an increase in D-Dimer may be a marker of severe viral infection progression due to inflammatory activation of the fibrinolytic system [41,42,43]. 

Endothelial cell damage and functional alteration due to SARS-CoV-2 have been demonstrated in many studies. According to a recent meta-analysis [44], convalescent COVID-19 patients may show an impaired endothelial function, as expressed by lower flow-mediated dilation (FMD) values than controls. The difference was significant by analyzing studies involving participants with no cardiovascular risk factors or previous coronary artery disease (CAD) history. Furthermore, the impairment was evident in both short-term and long-term follow-up studies. These findings are added to the growing body of scientific evidence supporting the role of endothelial dysfunction as a critical pathogenic mechanism of COVID-19 and its post-acute sequelae (PASC). It is now widely known that several residual clinical manifestations may persist beyond four weeks from symptom onset, even among patients experiencing mild or moderate disease. Endothelial dysfunction, hypercoagulability, and inflammation may still be detectable up to 1 year after recovery from COVID-19, as indicated by increased circulating levels of endothelin-1, intercellular adhesion molecule-1 (ICAM-1), IL-6, von Willebrand factor (vWF), D-dimer, and coagulation factor VIII.

In conclusion, impaired endothelial function can be documented in convalescent COVID-19 patients up to 1 year after infection, especially when residual clinical manifestations persist. Although the pathophysiological mechanisms underlying such persistent or delayed clinical manifestations of the convalescent phase are still a matter of study, several mechanisms have been proposed to have a role in the pathogenesis of PASC and long-term thrombotic complications, including the virus-induced downregulation of ACE2, inflammatory cytokines produced by activated leukocytes, endothelial dysfunction, hypercoagulability, and inflammation [45]. Adrenomedullin is a vascular permeability marker and is upregulated in systemic capillary leak syndrome in patients with severe COVID-19 [46,47,48,49]. Our study population’s adrenomedullin serum levels at admission were higher in GII patients (Table 4).

### 4.1. Krebs von den Lungen

Type II pneumocytes usually express KL-6; its increase is linked to pulmonary interstitial disease, including interstitial pneumonia, a frequent COVID-19 manifestation. Previous studies [50,51,52] have shown high KL-6 serum concentrations (>1000 U/mL) in severe COVID-19 patients, suggesting a role as a severity biomarker. In agreement with those statements, our study population demonstrated increased KL-6 levels, significantly higher in GII (μ_II_ = 1159 U/mL) with worse outcomes. In addition, KL-6 levels in our study population showed a positive correlation with CRP, NAg, and TSS, as verified by the Pearson correlation test (Table 4 and Table 5). 

### 4.2. SARS-CoV-2 Nucleocapsidic Antigen

Some interest in serum detection of NAg in COVID-19 patients was born in the first months of the COVID-19 pandemic, and it was considered a possible diagnostic tool. NAg serum dosage has some essential features as a diagnostic tool: (1) temporal advantage, since NAg can be detected before seroconversion; (2) high sensitivity (92%); 3) high specificity (96.84%). This simple and quick tool at hospital admission may speed up the identification of infected patients and lead to early insulation [53]. 

The present study focuses on the potential use of NAg as a severity biomarker of COVID-19, owing to its broad expression during the active viral replication phase and considering its intrinsic immunogenic power. Some studies highlighted higher levels of NAg in more severe forms of COVID-19 pneumonia admitted in ICU with a significant increase in inflammation markers as RCP [23]. 

We showed that patients with unfavorable outcomes have significantly increased levels of serum NAg at hospital admission (Table 4 and Table 5). SARS-CoV-2 leads to a systemic disease due to the broad expression of ACE-2 receptors in different tissues with a robust inflammatory activation. NAg can promote NLRP3 inflammasome hyperactivation that causes the release of many proinflammatory cytokines: IL-6, IL-1β, tumor necrosis factor α (TNF-α), interferon β (INF-β), and transforming growth factor β (TGF-β) [54,55]. This “cytokine storm”, demonstrated in our study by increased cytokines levels such as IL-6 and IL2-R, is part of the immune dysregulation and increased mortality that affected more severe patients [56]. 

In addition, NAg can inhibit interferon-mediated immunity in the first days of infection and the subsequent release of proinflammatory cytokines, including IL-6, IL-1β, and TNF-α [55].

This study also highlights the relation between serum NAg elevation and TSS radiological evaluation of COVID interstitial pneumonia: higher radiological scores were identified in patients with greater clinical severity and worse outcomes (Table 5). This link is also demonstrated between TSS and KL-6, a pulmonary damage biomarker [57]. 

More severe COVID patients in our study presented a higher number of complications during their hospital stay. Anemia, bleeding, acute kidney damage, dystonia, and arrhythmias were evident in many patients owing to the ability of the virus to damage several organs directly or indirectly, especially those with greater expression of the ACE receptor [58,59,60].

## 5. Conclusions

The data analysis confirmed that the levels of biomarkers, cytokines, and inflammation markers of our study population were elevated both upon admission to the hospital and during their stay. Patients with unfavorable outcomes (ICU stay and death) had statistically significant higher values.

In conclusion, high levels of NAg during the viremic phase of infection lead to a massive inflammation state and organ damage, as testified by the increase in inflammation markers, cytokines, and biomarkers. KL-6 is a highly specific pulmonary damage biomarker and, like NAg, has a critical correlation with biochemical markers and imaging.

We suggest that NAg evaluation at the hospital admission should be considered a biomarker of severe COVID-19 since it is an ascertained inflammation promoter leading to systemic multiorgan damage. Even if serum NAg cannot be regarded as a first-choice diagnostic tool for SARS-CoV-2 infection, it has high sensitivity, specificity, and positive predictive value (PPV). Its evaluation should be considered an accessory diagnostic tool, allowing quick and cheap identification of infected patients with an increased risk of severe COVID-19 development and ensuring admission in the adequate care setting. 

Nowadays, this tool could be even more helpful since, with the end of the pandemic, it is essential to identify those with severe disease requiring hospitalization instead of identifying infected subjects.

It would be helpful to continue collecting data to identify which aspects of NAg could help define severe COVID-19.

KL-6’s increased values in our study population affected by severe COVID-19 are consistent with data highlighted in the literature and confirm its role as a biomarker of interstitial lung damage and a predictor of the most severe clinical presentations. 

Even if the cut-off (>1000–1300 IU/mL) suggested by other studies to predict a severe prognosis is confirmed by our data, a decision value for KL-6 as a diagnostic tool was not unanimously recognized by the literature.

Several studies concerning KL-6 investigated its link with interstitial lung diseases (ILD) conducive to lung fibrosis, e.g., idiopathic pulmonary fibrosis (IPF), nonspecific interstitial pneumonia (NSIP), and connective tissue disease ILDs (CTD-ILD). COVID-19 has not yet been definitively and univocally associated with an evolution towards pulmonary fibrosis. However, results from this study can be merged within a framework suggesting that KL-6 could be a promising tool for confirming the diagnosis of acute disease and predicting its prognosis.

## Figures and Tables

**Table 1 diagnostics-14-00642-t001:** Baseline characteristics of the total population, GI, and GII.

Characteristics	Population (μ)	GI (μ_I_)	GII (μ_II_)	*p*-Value
AGE (years)	64.75	60.56	71.26	GII > GI 0.00001 *
SEX (F/M)	36/92	18/58	18/34	0.16

Notes: μ: mean population; GI: Group I; GII: Group II; *: statistically significant (*p* > 0.05).

**Table 2 diagnostics-14-00642-t002:** Blood count parameters of the total population, GI, and GII.

Blood Count (Unit of Measure)	Population (μ)	GI (μ_I_)	GII (μ_II_)	*p*-Value
RBC (×10^6^/microliter)	4.7 (nr 4.3–5.9)	4.8	4.5	GI > GII 0.01 *
Hb (g/dL)	13.6 (nr 13–17)	13.9	13.1	GI > GII 0.01 *
WBC (×10^6^/microliter)	11.18 (nr 4.0–10.5)	9.5	10.3	0.47
LYM (%)	10.2 (nr 20–45)	9.5	8.3	0.13
MON (%)	5.7 (nr 3.4–11)	6.1	5.1	GI > GII 0.01 *
NEU (%)	84.3 (nr 40–75)	83.7	86.0	GII > GI 0.04 *
EOS (%)	0.3 (nr 0.0–7-0)	0.37	0.24	Gi > GII 0.03 *
BAS (%)	0.18 (nr 0.0–1.5)	0.21	0.16	0.7
PLT (×10^3^/microliter)	232 (nr 150–400)	238	206	GI > GII 0.007 *

Notes: μ: mean population; GI: Group I; GII: Group II; nr: normal range; RBC: red blood cells; Hb: hemoglobin; WBC: white blood cells; LYM: lymphocytes; MON: monocytes; NEU: neutrophils; EOS: eosinophils; BAS: basophils; PLT: platelets; *: statistically significant (*p* > 0.05).

**Table 3 diagnostics-14-00642-t003:** Blood chemistries of the total population, GI, and GII.

Blood Chemistries (Unit of Measure)	Population (μ)	GI (μ_I_)	GII (μ_II_)	*p*-Value
ALT (U/L)	52.5 (nr 10–49)	54.2	30.4	GI > GII 0.00002 *
AST (U/L)	48.6 (nr 0–34)	42.34	47.9	0.17
Albumin (g/dL)	3.83 (nr 3.2–4.8)	3.95	3.66	GI > GII 0.009 *
Total Proteins (g/dL)	6.6 (nr 6.4–8.3)	6.5	6.2	GI > GII 0.008 *
Creatinine (mg/dL)	1.26 (nr 0.6–1.1)	0.98	1.67	GII > GI 0.0001 *
Azotemia (mg/dL)	65.3 (nr 10–50)	54.1	81.4	GII > GI 0.000007 *
Sodium (mEq/L)	138.5 (nr 135–145)	138.2	138.8	0.11
Potassium (mEq/L)	4.48 (nr 3.5–5.1)	4.28	4.53	GII > GI 0.03 *
Calcium (mg/dL)	8.82 (nr 8.8–10.4)	9.05	8.51	GI > GII 0.00007 *
Corrected Calcium (mg/dL)	9.04 (nr 8.8–10.4)	9.19	8.82	GI > GII 0.01 *
Glycemia (mg/dL)	139 (nr 74–106)	118	143	GII > GI 0.002 *
25-OH D_3_ (ng/mL)	21.52 (nr > 30)	22	18	GI > GII 0.03 *

Notes: μ: mean population; GI: Group I; GII: Group II; nr: normal range; ALT: alanine aminotransferase; AST: aspartate aminotransferase; 25-OH D_3_: 25-hydroxyvitamin D_3_; *: statistically significant (*p* > 0.05).

**Table 4 diagnostics-14-00642-t004:** Acute phase reactants and TSS of the total population, GI, and GII.

Laboratory and Radiological Markers (Unit of Measure)	Population (μ)	GI (μ_I_)	GII (μ_II_)	*p*-Value
ESR (mm/h)	58.4 (nr < 10)	50.9	72.2	GII > GI 0.005 *
RCP (mg/dL)	10.8 (nr < 1)	8.8	12.4	GII > GI 0.03 *
PCT (ng/mL)	0.41 (<0.05)	0.22	1.42	GII > GI 0.001 *
Adrenomedullin (ng/mL)	1.64 (nr < 1.01)	1.34	1.98	GII > GI 0.0002 *
D-Dimer (mg/dL)	462 (nr < 250)	230	818	GII > GI 0.000003 *
IL-6 (pg/mL)	58.5 (nr 0–6)	41.3	84.8	GII > GI 0.0001 *
IL-2R (U/I)	1516.9 (nr 150–600)	1316.3	1813.6	GII > GI 0.001 *
Fibrinogen (mg/dL)	573 (nr < 100)	567	580	0.6
Ferritin (ng/mL)	841 (nr 5–300)	1023	736	GI > GII 0.005 *
NAg (pg/mL)	137.96 (nr 0–1)	73.83	400.77	GII > GI 0.001 *
KL-6 (U/mL)	890 (nr 200–600)	670	1159	GII > GI 0.02 *
TSS	13	11	15	GII > GI 0.00001 *

Notes: μ: mean population; GI: Group I; GII: Group II; nr: normal range; ESR: erythrocyte sedimentation rate; RCP: C reactive protein; PCT: procalcitonin; IL-6: interleukin-6; IL-2R: interleukin-2 receptor; NAg: nucleocapsidic antigen; KL-6: Krebs von den Lungen-6; *: statistically significant (*p* > 0.05).

**Table 6 diagnostics-14-00642-t006:** Comorbidities prevalence among the total population, GI and GII.

Comorbidities	Population (Prevalence)	GI (Prevalence)	GII (Prevalence)	OR	χ^2^	*p*-Value
Anemia	64/128 (0.5)	27/76 (0.35)	37/52 (0.71)	4.48	15.76	<0.0001
Renal failure	30/128 (0.23)	3/76 (0.03)	27/52 (0.51)	26.28	39.60	<0.0001
Disiony	70/126 (0.51)	30/76 (0.39)	40/52 (0.76)	5.11	17.47	<0.0001

GI: Group I, GII: Group II; OR: odds ratio; χ^2^: chi-square test.

## Data Availability

Data are contained within the article.

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
