# Peer review of "The Role of SARS-CoV-2 Nucleocapsidic Antigen and Krebs von den Lungen 6 Serum Levels in Predicting COVID-19 Pneumonia Outcome"

_diagnostics, 2024, doi:10.3390/diagnostics14060642_

Round 1

Reviewer 1 Report

Comments and Suggestions for Authors

Coronavirus infection continues to be one of the health problems of all countries.  Given that some cases of the disease are still fatal, and some patients develop post-COVID syndrome, it is very important to determine how the infectious process will develop as early as possible. The authors of the reviewed article investigated the possibility of using the level of Krebs von den Lungen glycoprotein (KL-6) and the level of the N-protein of the SARS-Cov-2 virus (NAg) to predict the course and outcome of coronavirus infection. Several studies have previously shown that KL-6 may be such a predictor. As for the level of NAg of the coronavirus, such studies were conducted by the authors of the article for the first time. The article provides convincing evidence that the level of NAg of the coronavirus, as well as the level of KL-6, can serve as a predictor of the severity of the course of coronavirus infection. The methods used in this work are fully adequate to the tasks set. Tables and graphs presented by authors are informative and fully reflect the results obtained during the research. The article is written in good English and does not need to be corrected. The article can be published without major changes.

Author Response

I truly appreciate your dedication and commitment to the peer-review process, which has helped me refine my research and present it more effectively. Your expertise and input have been instrumental in strengthening my work, and I am grateful for your contribution.

Reviewer 2 Report

Comments and Suggestions for Authors

Zamparelli & al. examine the suitability for SARS-CoV-2 diagnosis, and especially for the severity and outcome of pneumonia, of two serum components, the viral nucleocapsid antigen NAg and the human cell-derived KL-6. This task overall is accomplished in the manuscript. Among its strengths are excellent English and overall medical writing. While the literature on the latter marker is cited satisfactorily, that on the former could be more comprehensive. I merely have some minor comments for fine-tuning of the text.

Minor comments

(1) The authors could attempt to correlate the presence of the two markers with the actual _duration_ of SARS-CoV-2 infection or clinical picture. At what chronological stage of infection/illness were the two markers detectable – and at which levels – in the patients’ sera? This question comes to mind e.g. on line 200, where it now reads “in GII, who had worse outcomes”. How about durations? 

(2) In Figure 1 (and 2), upon visual inspection (cf. statistical values), most of the NAg correlations actually look quite modest. The authors might want to comment on this, perhaps even in the text. 

(3) Line 212, “PCR”, please define.  

(4) The term “COVSIE (population)” occurs often throughout the manuscript (after line 102), yet without apparent purpose and precise definition. Which study population(s) does this term refer to, include and exclude? Do we need this term at all?

(5) The manuscript title including the viral antigen could include also the other marker (KL-6).

(6) Line 65, “encapsulating” – a word I'm not familiar with?

(7) Has the Lumipulse test been validated for serum use (vs. nasal)?

Comments on the Quality of English Language

Please see my Comments for Authors.

Author Response

I would like to express my sincerest appreciation for the dedication and commitment you have shown towards the peer-review process. Thank you for your efforts and the time you have invested in this process.

I have taken note of your comments and will provide a detailed response addressing each point individually.

1) The present study evaluated two specific markers in patients, taking measurements at the time of admission and subsequently monitoring them until discharge or in-hospital death/ admission to the intensive care unit (ICU). Although not explicitly mentioned, the correlation between high levels of NAg and KL-6 biomarkers and ICU stay underlies extensions in the hospitalization length.

(2) The following sentences were added in line 202: Despite the modest correlation between Nag and the primary laboratory and/or radiological biomarkers of inflammation, it is noteworthy that there exists a positive correlation between NAg KL-6. These two biomarkers have shown potential in assessing the severity of lung involvement. 

(3) The acronym PCR was incorrect and has been replaced with the appropriate term CRP.

(4) As suggested, we have removed the term COVSIE, which refers to the population included in the study with detectable serum levels of SARS-CoV-2 NAg > 1 pg/mL.

5) As suggested, the manuscript title was changed to ‘The Role of SARS-CoV-2 Nucleocapsidic Antigen and Krebs von den Lungen 6 serum levels in predicting COVID-19 pneumonia outcome’.

(6) In order to be clearer, we used the word "enveloping" as a substitute for another word.

(7) We performed an assessment of the presence of NAg in the serum. However, it should be noted that the method used was not specifically validated for serum samples, but rather for the swab universal viral transport fluid (UVT) that is provided for nasal and pharyngeal swab smears. Upon detecting the NAg presence in the serum, we conducted additional research to assess the accuracy of the data, using a molecular method known as Polymerase Chain Reaction (PCR) on a series of samples and finding a strong correlation between the level of viremia and the level of NAg as measured by our method (unreported data). We have further validated the efficacy of our method by utilizing healthy subjects' serum to fluidize the lyophilized NAg provided by Fujirebio. This approach enabled us to prepare control samples at varying concentrations, which were subsequently utilized to obtain relative correlation findings with viremia.

To be clearer to the reader, we added the following sentence at line 118 with the relative reference (23): ‘In a previous study, we demonstrated the efficacy of the Lumipulse® test for serum utilization by examining the correlation between SARS-CoV-2 NAg and RNA serum levels’.
